# HiCBridge: Resolution Enhancement of Hi-C Data Using Direct Diffusion Bridge

## Abstract

Hi-C analysis provides valuable insights into the spatial organization of chromatin, which affects many aspects of genomic processes. However, the usefulness of Hi-C is hindered by its resolution limitations owing to the sequencing cost. Here, we propose **Hi-C** enhancement using Direct Diffusion **Bridge** (HiCBridge) that learns transformation from low-resolution Hi-C data to high-resolution ones using direct diffusion bridge (DDB). Instead of relying on standard supervised feed-forward networks and GANs, which often produces overly smooth textures or falls into mode collapsing, the main idea of HiCBridge is building a diffusion process, by directly bridging the low and high-resolution Hi-C data. Furthermore, to make our model applicable in real-world situations, we further train our model by increasing the variation of the real-world data with diffusion model-based data augmentation. We demonstrate that our model can be used to improve downstream analyses such as three-dimensional structure matching, loop position reconstruction, and recovery of biologically significant contact domain boundaries. Experimental results confirm that HiCBridge surpasses existing deep learning-based models on standard vision metrics, and exhibits strong reproducibility in Hi-C analysis of human cells.

## 1 Introduction

High-throughput chromosome conformation capture sequencing (Hi-C) is a powerful genomic and epigenomic technique that offers information about the three-dimensional (3D) structure of the genome (Lieberman-Aiden et al., 2009). While previous techniques such as 3C (Dekker et al., 2002), 4C (Zhao et al., 2006), and 5C (Dostie et al., 2006) have been proposed to capture chromosome conformation, the Hi-C method provides the advantage of capturing all possible contacts within and between chromosomes (Varoquaux et al., 2014). By measuring the frequency of paired chromatin interactions, Hi-C analysis enables the identification of important conformational features of the genome, including A/B compartments (Lieberman-Aiden et al., 2009), gene regulatory mechanisms (Rao et al., 2014; Wang et al., 2016; Schmitt et al., 2016), topology associated domains (TADs) (Dixon et al., 2012), and chromatin loops (Rao et al., 2014).

To effectively utilize Hi-C data, the sequence read counts need to be converted into a matrix of contacts. These matrices are indexed by rows and columns corresponding to genomic regions and are partitioned into fixed bin sizes (Lajoie et al., 2015). Therefore, in the realm of Hi-C data analysis, the 'resolution' or bin size is a pivotal factor that influences the outcome of various analyses, like identifying regulatory regions or boundary regions in the genome. Achieving kilobase-scale resolution in Hi-C data has thus become increasingly important for accurately elucidating 3D genome structures. However, the reality of deep sequencing costs means that many existing Hi-C datasets are of lower resolution, typically around 25 or 40 kb. This is because linear resolution increases demands quadratic increase of sequence reads, making it a challenging and expensive endeavor (Schmitt et al., 2016). Nonetheless, an insufficient number of reads leads to noisy and structurally uninformative data (Jin et al., 2013; Filippova et al., 2014; Dixon et al., 2015; Durand et al., 2016).

To address this challenge, previous researchers have employed deep learning-based methods to convert low-resolution data into high-resolution Hi-C data. HiCPlus (Zhang et al., 2018) and HiCNN (Liu & Wang, 2019) leverage convolutional layers and optimize the mean squared error (MSE) loss to map low-resolution to high-resolution Hi-C matrices. Other methods, including

HiCSR (Dimmick, 2020), DeepHiC (Hong et al., 2020), and VEHiCLE (Highsmith & Cheng, 2021), utilize generative adversarial neural networks (GANs) (Goodfellow et al., 2020) and employ additional loss functions to produce Hi-C matrices with sharper and more realistic features. Although these previous studies have demonstrated high performance in improving Hi-C resolution, they suffered from certain drawbacks. For example, models relying on regression and the MSE loss tend to produce images with overly smooth textures (Mathieu et al., 2015), and GAN-based models can suffer from unstable learning or mode collapse (Goodfellow et al., 2020). Adding explicit losses to improve performance may inadvertently result in the learning of unwanted natural image textures (Dimmick, 2020). Since the synthesized low-resolution data for training are different from the distribution of the real ones, inference to the real data results in poor performance (Murtaza et al., 2022).

To address these, here we propose an Hi-C enhancement using Direct Diffusion Bridge (HiCBridge) model that enhances Hi-C resolution in a simple yet effective manner. Specifically, we employ the recent concept of Direct Diffusion Bridge (DDB) (Chung et al., 2023) which was inspired by Inversion by direct iteration (InDi) (Delbracio & Milanfar, 2023) that iteratively denoises the data through diffusion bridge. Specifically, by breaking down the mapping process into smaller steps along the diffusion bridge, our model effectively learns the mapping from low-resolution to high-resolution Hi-C data without falling into mode collapsing as seen in GANs or texture blurring common in standard supervised deep learning. On the other hand, to create a model applicable to real-world scenarios, we need to train HiCBridge using genuine data to prevent the model from losing its generalization ability. To mitigate this potential risk, another important contribution of this paper is a diffusion model (Ho et al., 2020) that generates new low-resolution data from high-resolution one. This approach diversifies the data and enhances the generalization performance of HiCBridge, which we denote as HiCBridge$^+$.

Comparative experiments with previous deep learning-based models demonstrate that HiCBridge$^+$ achieves the state-of-the-art performance on several standard vision metrics. Additionally, HiCBridge and HiCBridge$^+$ outperforms other models on various biological metrics, providing validation of Hi-C data reproducibility. Moreover, our framework excels in downstream tasks that measure recovery of structural information. Finally, we verify the suitability of our model for real-world Hi-C analysis by training the model on real-world low-resolution Hi-C data and comparing the results with reconstructed high-resolution Hi-C data across different cell types and resolutions.

Our contributions can be summarized as follows:

- Using direct diffusion bridge, we propose an HiCBridge model that shows high performance without any additional explicit losses.
- To enhance its generalization performance for real-world data, we propose HiCBridge$^+$, by finding the optimal combination of using diffusion model to enrich HiCBridge's data.
- The proposed model shows the highest performance on metrics on standard vision metrics, various biological metrics, and downstream tasks across different cell types and resolutions.

## 2 BACKGROUNDS

**Hi-C data acquisition and resolution.** Hi-C data offer profound insights into the 3D structural information of chromatin within cell nuclei (Lieberman-Aiden et al., 2009). The acquisition of Hi-C involves sequencing spatially proximate DNA fragments to reveal long-range interactions across an entire genome. Raw Hi-C data represent pairwise reads counted at corresponding locations on chromosome, which are then converted into 2D contact matrices indicating the contact frequency of each bin. Details of pre-processing we used are provided in the Appendix A.1.

The resolution of Hi-C data depends on the bin size of the Hi-C contact matrix. Similar to pixel size in images, shorter bins (e.g., 10kb) yield more detailed chromosome contact information (Liu & Wang, 2019), while coarser resolutions with larger bins (e.g., 1Mb) provide a broader perspective. The resolution of experimentally obtained Hi-C data is proportional to the number of read pairs, and insufficient read coverage indicates sparse contact frequency. A $n$ fold increase in resolution requires $n^2$ read pairs, resulting in higher costs for Hi-C experiments. Therefore, the low-resolution Hi-C data in the standard binning size produces noisy contact maps due to the low sequence reads.

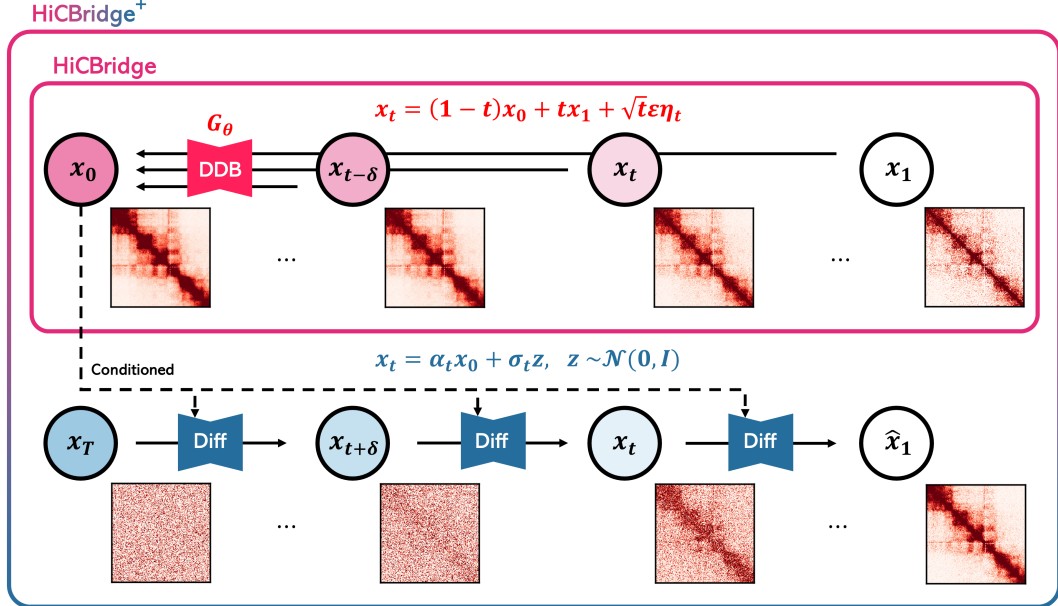

Figure 1: Model pipeline overview. HiCBridge framework produces high-resolution data from corrupted data by applying noising procedure Eq. (6) and denoising through $G_\theta(\boldsymbol{x}_t)$. To increase the diversity in the Hi-C dataset, we further train a diffusion model to generate low-resolution data by conditioning on high-resolution data. HiCBridge$^+$ is the HiCBridge trained with diffusion model-based data augmentation.

**Existing deep learning approaches for Hi-C enhancement.** Previous deep learning-based methods to enhance Hi-C resolution can be broadly categorized into two types based on their objectives: those employing the Mean Squared Error (MSE) loss and those using Generative Adversarial Network (GAN) loss. HiCPlus and HiCNN optimize the MSE loss using convolutional neural networks to enhance downsampled low-resolution Hi-C matrices. However, regressing to high-resolution through the MSE loss often results in blurry outputs. With the advent of GANs, models such as HiCSR, DeepHiC, and VEHiCLE utilize GAN loss to generate more realistic high-resolution data from low-resolution counterparts. More specifically, the HiCSR model employs a task-specific autoencoder to compare the differences between features in the latent space. DeepHiC incorporates a perceptual loss (Johnson et al., 2016) and total variation loss, while VEHiCLE employs a variational autoencoder (VAE) (Kingma & Welling, 2013) to extract intrinsic important features from Hi-C data and utilizes an explicit loss based on the TAD insulation score. However, GAN-based models are susceptible to mode collapse, and models relying on explicit losses may introduce artifacts inconsistent with the Hi-C data (Dimmick, 2020).

In Fig. 2, we highlight the limitations of models trained with traditional MSE and GAN approaches. HiCPlus, an MSE-based model, fails to capture existing TAD regions present in high-resolution data, whereas the GAN-based model, HiCSCR, exhibits outliers in locations where they should not be. Those artifacts could potentially lead to misinterpretation of the reconstructed Hi-C data.

In addition, with the exception of VEHiCLE, previous deep learning models were trained using synthetic low-resolution data downsampled to a specific resolution (e.g., 1/16 read counts) from high-resolution raw Hi-C data, not contact matrices. A study by Murtaza et al. (2022) demonstrated differences in the distribution of real low-resolution data compared to synthetic counterparts, potentially resulting in sub-optimal performance in real-world situation.

**Direct diffusion bridge.** Here we briefly review Direct Diffusion Bridge (DDB) (Chung et al., 2023). Consider the case where we can sample $\boldsymbol{x}_0 := \boldsymbol{x} \sim p(\boldsymbol{x})$, and $\boldsymbol{x}_1 := \boldsymbol{y} \sim p(\boldsymbol{y}|\boldsymbol{x})$, i.e. paired data for training. This diffusion bridge introduces a *continual* degradation process by taking a convex combination of $(\boldsymbol{x}_0, \boldsymbol{x}_1)$, starting from the clean image at $t = 0$ to maximal degradation at $t = 1$, with additional stochasticity induced by the noise component $\sigma_t$. This can be represented as

$$\boldsymbol{x}_t = (1 - \alpha_t)\boldsymbol{x}_0 + \alpha_t\boldsymbol{x}_1 + \sigma_t\boldsymbol{z}, \ \boldsymbol{z} \sim \mathcal{N}(0, \boldsymbol{I}), \quad (1)$$

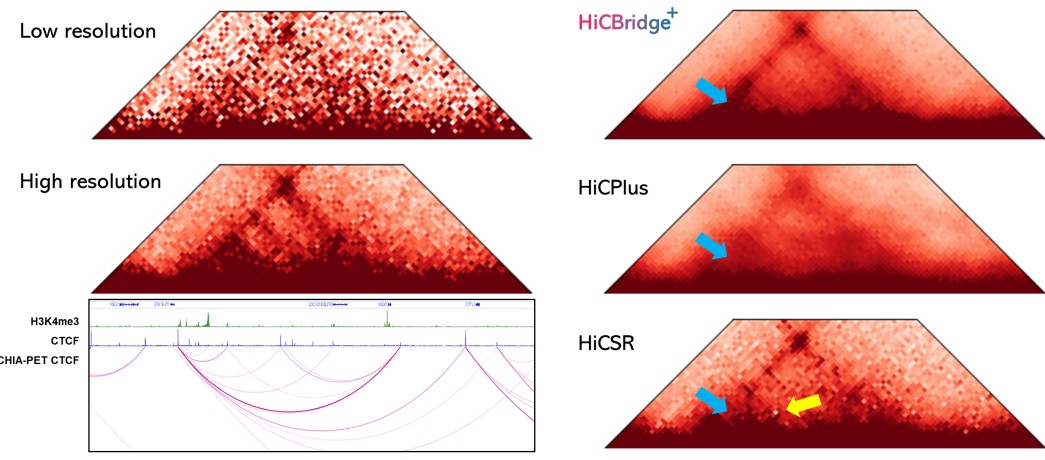

Figure 2: Comparison of Hi-C data and enhancement results on GM12878 chromosome 5 4.4Mb-5.3Mb. Blue arrows indicate TAD regions that should be present in existing high-resolution data. Yellow arrow indicates outliers that are not present in existing data. Our model showcases conservative reconstruction, preserving structural information without introducing artifacts that could potentially lead to misinterpretation of the reconstructed Hi-C data, whereas the GAN-based model, HiCSCR, exhibits outliers in locations where they should not be.

where $\alpha_t, \sigma_t^2$ are time-dependent parameters. Our goal is to train a time-dependent neural network that maps any $\boldsymbol{x}_t$ to $\boldsymbol{x}_0$ that recovers the clean image. The training objective is

$$\min_\theta \mathbb{E}_{\boldsymbol{x}_t \sim p(\boldsymbol{x}_t|\boldsymbol{x}_0), \boldsymbol{x}_0 \sim p(\boldsymbol{x}_0), t \sim U(0,1)}[\|G_\theta(\boldsymbol{x}_t) - \boldsymbol{x}_0\|_2^2], \tag{2}$$

which is equivalent to the denoising score matching (DSM) (Hyvärinen & Dayan, 2005):

$$\min_\theta \mathbb{E}_{\boldsymbol{y} \sim p(\boldsymbol{y}|\boldsymbol{x}), \boldsymbol{x} \sim p(\boldsymbol{x}), t \sim U(0,1)} \left[ \|\boldsymbol{s}_\theta(\boldsymbol{x}_t) - \frac{\boldsymbol{x}_t - \boldsymbol{x}_0}{\gamma_t}\|_2^2 \right], \tag{3}$$

Once the network is trained, we can reconstruct $\boldsymbol{x}_0$ starting from $\boldsymbol{x}_1$ by, for example, using DDPM ancestral sampling (Ho et al., 2020), where the posterior for $s < t$ reads

$$p(\boldsymbol{x}_s|\boldsymbol{x}_0, \boldsymbol{x}_t) = \mathcal{N}(\boldsymbol{x}_s; (1 - \alpha_{s|t}^2)\boldsymbol{x}_0 + \alpha_{s|t}^2 \boldsymbol{x}_t, \sigma_{s|t}^2 \boldsymbol{I}), \tag{4}$$

with $\alpha_{s|t}^2 := \frac{\gamma_s^2}{\gamma_t^2}$, $\sigma_{s|t}^2 := \frac{(\gamma_t^2 - \gamma_s^2)\gamma_s^2}{\gamma_t^2}$. At inference, $\boldsymbol{x}_0$ is replaced with a neural network-estimated $\hat{\boldsymbol{x}}_{0|t}$ to yield $\boldsymbol{x}_s \sim p(\boldsymbol{x}_s|\hat{\boldsymbol{x}}_{0|t}, \boldsymbol{x}_t)$ from $G_{\theta^*}$, where we simply denote the trained networks as $G_{\theta^*}$ even if it is parameterized otherwise.

## 3 METHODS

### 3.1 HICBRIDGE

The conventional regression models based on the Mean Squared Error (MSE) loss causes the model to regress to the mean in the target domain, resulting in blurring and losing details. To mitigate this problem, we propose Hi-C enhancement using Direct Diffusion Bridge (HiCBridge) using direct diffusion bridge (DDB) which breaks down the regression into smaller steps.

Specifically, we establish a stochastic relationship between the contact matrix from high-resolution Hi-C data, $\boldsymbol{x}_0 \in \mathbb{R}^{H \times W}$, and its low-resolution counterpart, $\boldsymbol{x}_1 \in \mathbb{R}^{H \times W}$ as

$$\boldsymbol{x}_1 = \mathbf{A}(\boldsymbol{x}_0) \sim p(\boldsymbol{x}_1|\boldsymbol{x}_0), \tag{5}$$

where $\mathbf{A}$ represents a *stochastic, non-linear and unknown* operator. We can conceptualize $\mathbf{A}$ as generating a contact matrix of low-resolution data from corresponding high-resolution data, following the distribution $p(\boldsymbol{x}_1|\boldsymbol{x}_0)$. Unfortunately, unlike conventional image restoration (Richardson, 1972), we do not possess prior knowledge about this operator. Moreover, $\mathbf{A}$ possesses stochastic characteristics, making a one-to-one matching of paired data unattainable.

Given the stochastic nature of the mapping, our goal is to exploit the direct diffusion bridge model to link the paired data $\boldsymbol{x}_0$ and $\boldsymbol{x}_1$ as in (1). In particular, a study by Chung et al. (2023) show that we can reformulate (1) as

$$\boldsymbol{x}_t = (1-t)\boldsymbol{x}_0 + t\boldsymbol{x}_1 + \sqrt{t}\epsilon\eta_t, \qquad (6)$$

when $\alpha_t = t$ and $\sigma_t = \sqrt{t}\epsilon\eta_t$. Here, $\eta_t$ represents the standard Brownian motion that accounts for the stochastic nature of the contact matrix from low-resolution Hi-C data, and $\epsilon$ is a small constant that controls the strength of the noise. Then, in line with (4), we can obtain the denoised data by a small step size $\delta$ by

$$\boldsymbol{x}_{t-\delta} = \frac{\delta}{t}G_{\theta^*}(\boldsymbol{x}_t) + \left(1 - \frac{\delta}{t}\right)\boldsymbol{x}_t. \qquad (7)$$

Here, we choose $\delta = \frac{1}{1000}$ at training and $\delta = 1$ at inference. Specifically, as shown in Fig. 1, starting from the noisy contact matrix $\boldsymbol{x}_t$ from low-resolution Hi-C data, our method estimates the corresponding estimates of the clean contact map through $G_{\theta^*}(\boldsymbol{x}_t)$. With the estimate $G_{\theta^*}(\boldsymbol{x}_t)$, Eq. (7) generates the cleaner version of the contact matrix $\boldsymbol{x}_{t-\delta}$ compared to $\boldsymbol{x}_t$. In particular, by using $\delta = 1$ at inference, we can arrive at the final estimate of the clean contact matrix in just one step, which makes the algorithm very fast.

## 3.2 HiCBridge$^+$: Diffusion-based Low-Resolution Data Augmentation

In order to enhance the practicality and applicability of our model, we train models using data obtained from actual experiments. Unfortunately, the insufficiency of Hi-C data can lead to overfitting of the model. As discussed in Section 3.1, the mapping of the Hi-C contact matrix is not precisely known, necessitating multiple experiments to collect sufficient Hi-C data to augment the dataset. However, this data collection process is time-consuming and expensive. To overcome this challenge, we train a diffusion model to sample the data from the unknown operator $\mathbf{A}$.

Specifically, we train a DDPM model (Ho et al., 2020) to generate new low-resolution Hi-C data corresponding to the high-resolution ones. During the training process, we concatenated the high-resolution data to the input for conditioning. Our diffusion model $\epsilon_\theta$ is trained to predict the various level of noises from corrupted input using Algorithm 1.

Using this trained diffusion model, we generate new low-resolution Hi-C data that align with their original high-resolution counterparts by Algorithm 2. Although the learned denoising function can be used to generate new low-resolution data from Gaussian noise, we employ the method introduced by Meng et al. (2021) and Chung et al. (2022) to accelerate generation speed. Specifically, we perturb the high-resolution data at $t = 0.5$ and then denoise them over the half iterations, conditioning the high-resolution data at each step. Subsequently, we train a new HiCBridge using an augmented Hi-C dataset, combining the generative power of diffusion with the regression capability of DDB. We call this model as HiCBridge$^+$. The pipeline of HiCBridge$^+$ is shown in Fig. 1. We visualized generated low-resolution using diffusion model in Appendix F.

| **Algorithm 1** Training | **Algorithm 2** Data augmentation |
|---|---|
| **Require:** $\boldsymbol{x}_1, \{\sqrt{\bar{\alpha}_t}\}_{t\in[0,1]}$ | **Require:** $\boldsymbol{x}_0, \{\sqrt{\bar{\alpha}_t}, \sigma_t\}_{t\in[0,1]}, \epsilon_\theta^{(t)}, \delta$ |
| $\quad$ **repeat** | $\quad t \leftarrow 0.5$ |
| $\quad\quad \epsilon \sim \mathcal{N}(0, \boldsymbol{I})$ | $\quad \epsilon \sim \mathcal{N}(0, \boldsymbol{I})$ |
| $\quad\quad t \sim U(0,1)$ | $\quad \boldsymbol{x}_t \leftarrow \sqrt{\bar{\alpha}_t}\boldsymbol{x}_0 + \sqrt{1-\bar{\alpha}_t}\epsilon$ |
| $\quad\quad \boldsymbol{x}_t \leftarrow \sqrt{\bar{\alpha}_t}\boldsymbol{x}_1 + \sqrt{1-\bar{\alpha}_t}\epsilon$ | $\quad$ **while** $t > 0$ **do** |
| $\quad\quad L = \\|\epsilon_\theta^{(t)}(\boldsymbol{x}_t\|\boldsymbol{x}_1) - \epsilon\\|^2$ | $\quad\quad z \sim \mathcal{N}(0, \boldsymbol{I})$ |
| $\quad\quad$ Take gradient decent step on | $\quad\quad \boldsymbol{x}_{t-\delta} = \frac{1}{\sqrt{\alpha_t}}\left(\boldsymbol{x}_t - \frac{1-\alpha_t}{\sqrt{1-\bar{\alpha}_t}}\epsilon_\theta^{(t)}(\boldsymbol{x}_t\|\boldsymbol{x}_0)\right) + \sigma_t z$ |
| $\quad L$ for $\theta$ | $\quad\quad t \leftarrow t - \delta$ |
| $\quad$ **until** converged | **end while** |

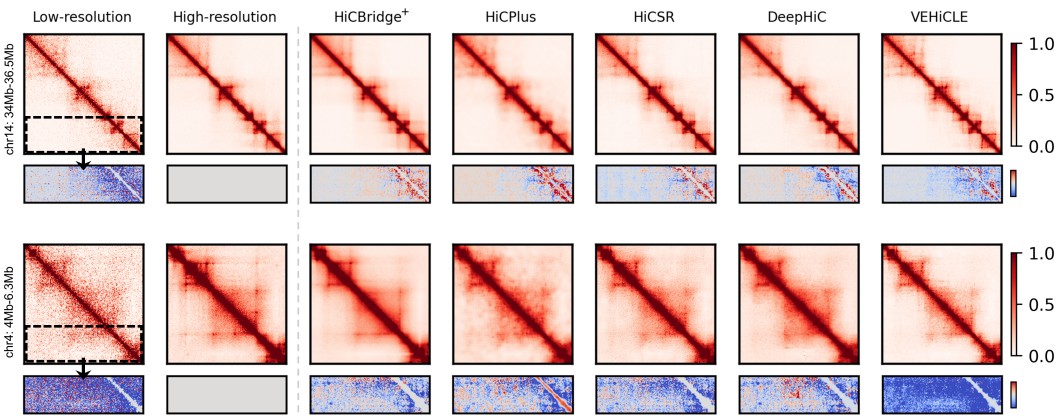

Figure 3: Comparison of Hi-C data and enhancement results by each model. The left two columns represent the low-resolution and high-resolution data of the actual GM12878 Hi-C data. The odd rows represent the Hi-C data and the even rows represent the difference from the high-resolution data of the corresponding genome region.

# 4 EXPERIMENTAL RESULTS

## 4.1 EVALUATION METHODS

To ensure the evaluation of our proposed method for Hi-C resolution enhancement, we trained all models on the real Hi-C data and utilized standard vision metrics commonly employed in deep learning-based models. These metrics include the Pearson correlation coefficient (PCC), Spearmans correlation coefficient (SCC), peak signal-to-noise ratio (PSNR), structural similarity index metric (SSIM), mean squared error (MSE), and signal-to-noise ratio (SNR). However, Yang et al. (2017) indicated that these correlation-based metrics do not fully capture the reproducibility of Hi-C data. To address this, we incorporated two Hi-C specific similarity metrics, GenomeDISCO (Ursu et al., 2018) and HiCRep (Yang et al., 2017). GenomeDISCO provides a concordance score by employing graph random walks to denoise contact maps, and HiCRep calculates a stratum-adjusted correlation coefficient that accounts for distance-dependency weights by stratifying Hi-C data.

Furthermore, we conducted three downstream tasks to assess the reconstruction of structural information in Hi-C data. The overview of downstream analysis is shown in Fig. 4. Firstly, we compared the 3D structure of chromatin based on the template modeling score (TM-Score) (Zhang & Skolnick, 2004) between the reconstructed Hi-C data and the high-resolution reference. We obtained the 3D structure of chromatin using 3DMax (Oluwadare et al., 2018) and compared the average TM-score. Secondly, we assessed the accuracy of TAD positioning by comparing the Insulation score (Crane et al., 2015). Since TAD represents a region with strong physical interactions, pinpointing the location of TADs has biological significance. We computed the TAD insulation vector by sliding the window and obtained the $L_2$ difference from the high-resolution data like in previous work (Highsmith & Cheng, 2021). Lastly, we validated the models' capability to accurately reconstruct chromatin loop positions by comparing the Jaccard Index of the reconstructed loops. We used Fit-Hi-C (Ay et al., 2014) to identify important loop positions and assessed how much the enhanced data shared loop positions with the high-resolution reference.

## 4.2 COMPARATIVE EVALUATION IN TERMS OF VISUAL METRICS

We conducted a comprehensive comparison between the HiCBridge, HiCBridge[+] and several deep learning-based models, namely HiCPlus, HiCSR, DeepHiC, and VEHiCLE. The evaluation was based on standard vision metrics, using Hi-C data from chromosomes 4, 14, 16, and 20 of GM12878 cells. As shown in Table 1, our HiCBridge[+] model demonstrated superior performance across all metrics, with significant margins. When comparing results from other previous models, incorporating additional loss functions tended to improve the vision metrics. The actual Hi-C data used for testing and the reconstructed Hi-C data of each model are displayed in Fig. 3. Note that HiCPlus, which optimized the MSE loss, produced blurry output, and HiCSR and DeepHiC, constrained by inadequately large image sizes, exhibited grid-like artifacts. And VEHiCLE, utilizing explicit losses,

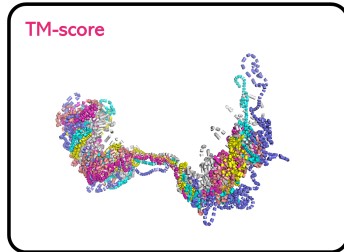 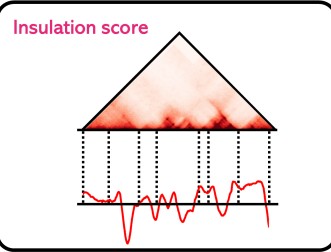 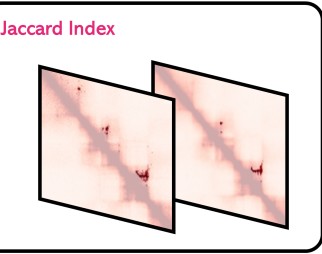

Figure 4: Illustration of downstream analyses of reconstructed Hi-C data. TM-Score assesses the structural similarity, and Insulation score helps identify TAD boundaries by calculating average interaction frequency within sliding windows. Jaccard Index measures the accuracy of reconstructed significant chromatin loops.

Table 1: Quantitative evaluation of average standard vision metrics on GM12878 test Hi-C data. $^+$ indicates that diffusion augmentation is applied. **Bold**: best, underline: second best.

| Method | PCC | SCC | PSNR | SSIM | MSE (↓) | SNR |
|---|---|---|---|---|---|---|
| Low-resolution | 0.854 | 0.667 | 18.02 | 0.352 | 0.02091 | 326.4 |
| HiCPlus (Zhang et al., 2018) | 0.906 | 0.742 | 20.13 | 0.554 | 0.01107 | 413.9 |
| HiCSR (Dimmick, 2020) | 0.909 | 0.750 | 20.13 | 0.523 | 0.01123 | 413.9 |
| DeepHiC (Hong et al., 2020) | 0.909 | 0.730 | 20.24 | 0.542 | 0.01083 | 419.4 |
| VEHiCLE Highsmith & Cheng (2021) | 0.968 | 0.886 | 24.46 | 0.596 | 0.00582 | 694.2 |
| HiCBridge (ours) | 0.977 | 0.911 | 26.86 | 0.671 | 0.00294 | 911.1 |
| HiCBridge$^+$ (ours) | **0.978** | **0.912** | **27.05** | **0.676** | **0.00278** | **930.9** |

Table 2: Comparative results of HiCRep and GenomeDISCO on GM12878 cell line. 'chr' represents chromosome and $^+$ indicates diffusion augmentation is applied. **Bold**: best, underline: second best.

| Method | HiCRep | | | | GenomeDISCO | | | |
|---|---|---|---|---|---|---|---|---|
| | chr 4 | chr 14 | chr 16 | chr 20 | chr 4 | chr 14 | chr 16 | chr 20 |
| Low-resolution | 0.917 | 0.950 | 0.948 | 0.957 | 0.924 | 0.926 | 0.913 | 0.921 |
| HiCPlus (Zhang et al., 2018) | 0.943 | 0.966 | 0.960 | 0.966 | 0.953 | 0.944 | 0.911 | 0.915 |
| HiCSR (Dimmick, 2020) | 0.957 | 0.966 | 0.959 | 0.959 | 0.935 | 0.873 | 0.852 | 0.831 |
| DeepHiC (Hong et al., 2020) | 0.958 | 0.971 | 0.967 | 0.968 | 0.960 | 0.957 | 0.939 | 0.946 |
| VEHiCLE (Highsmith & Cheng, 2021) | 0.919 | 0.956 | 0.954 | 0.964 | 0.623 | 0.753 | 0.799 | 0.846 |
| HiCBridge (ours) | 0.968 | 0.979 | 0.974 | 0.975 | **0.965** | **0.961** | **0.946** | 0.939 |
| HiCBridge$^+$ (ours) | **0.984** | **0.987** | **0.982** | **0.987** | 0.947 | 0.946 | 0.931 | **0.947** |

resulted in unintended distributional shifting effects in the reconstructed Hi-C data. Our model out-performed all others without requiring additional loss functions. Furthermore, we observed that augmenting the data with a diffusion model resulted in even higher performance.

### 4.3 COMPARATIVE EVALUATION IN TERMS OF HI-C REPRODUCIBILITY

We employed reproducibility metrics specific to Hi-C data, namely HiCRep and GenomeDISCO, comparing with high-resolution Hi-C data. We used the same data as for the vision metric comparison and applied off-the-shelf parameters for the reproducibility assessment. As presented in Table 2, our model achieved the highest Hi-C reproducibility performance. Interestingly, despite showing high vision metric results, VEHiCLE performed poorly on GenomeDISCO and even lower than the low-resolution Hi-C data. These results are in accordance with the results of Murtaza et al. (2022).

Table 3: Comparative downstream results of average TM-Score, Insulation score (IS), and Jaccard Index on GM12878 test Hi-C data. $^+$ indicates that diffusion augmentation is applied. **Bold**: best, underline: second best.

| Method | TM-Score | IS ($\downarrow$) | Jaccard Index |
|---|---|---|---|
| Low-resolution | 0.454 | 4.221 | 0.086 |
| HiCPlus (Zhang et al., 2018) | 0.802 | 3.209 | 0.316 |
| HiCSR (Dimmick, 2020) | 0.708 | 2.986 | 0.440 |
| DeepHiC (Hong et al., 2020) | 0.509 | 3.161 | 0.432 |
| VEHiCLE (Highsmith & Cheng, 2021) | 0.506 | 3.743 | 0.425 |
| HiCBridge (ours) | **0.821** | 2.627 | 0.447 |
| HiCBridge$^+$ (ours) | 0.792 | **2.316** | **0.463** |

## 4.4 ABILITY TO RECOVER STRUCTURAL FEATURES OF CHROMOSOMES

To verify how successfully the reconstructed Hi-C data recovered the structural information of the high-resolution data, we evaluated the TM-Score, TAD Insulation score, and Jaccard Index of the reconstructed loops. In the Table 3, our model achieved the best performance on all of downstream analyses. Our model not only recovers 3D information of high-resolution Hi-C data and exhibits minimal differences in reconstructed TAD boundaries but also restores significant chromatin loop.

Table 4: The results of in different resolutions (1/16, 1/50 and 1/100) on GM12878 chromosome 20 Hi-C data. 'IS' represents Insulation score and $^+$ indicates that diffusion augmentation is applied. **Bold**: best, underline: second best.

| Method | 1/16 | | | 1/50 | | | 1/100 | | |
|---|---|---|---|---|---|---|---|---|---|
| | SSIM | HiCRep | IS ($\downarrow$) | SSIM | HiCRep | IS ($\downarrow$) | SSIM | HiCRep | IS ($\downarrow$) |
| Low-resolution | 0.425 | 0.989 | 1.938 | 0.235 | 0.972 | 3.436 | 0.153 | 0.951 | 4.921 |
| HiCPlus (Zhang et al., 2018) | 0.667 | 0.846 | 2.205 | 0.628 | 0.828 | 3.397 | 0.571 | 0.811 | 4.461 |
| HiCSR (Dimmick, 2020) | 0.621 | 0.859 | 2.103 | 0.531 | 0.851 | 3.392 | 0.476 | 0.835 | 4.406 |
| DeepHiC (Hong et al., 2020) | 0.641 | 0.854 | 2.163 | 0.582 | 0.845 | 3.655 | 0.508 | 0.829 | 5.779 |
| VEHiCLE (Highsmith & Cheng, 2021) | 0.750 | 0.588 | 2.003 | 0.628 | 0.852 | 3.053 | 0.525 | 0.819 | **4.124** |
| HiCBridge (ours) | 0.787 | 0.988 | 2.074 | **0.757** | **0.976** | 3.053 | **0.717** | **0.963** | 4.416 |
| HiCBridge$^+$ (ours) | **0.796** | **0.990** | **1.837** | 0.751 | 0.970 | **2.944** | 0.643 | 0.944 | 4.143 |

Table 5: The average results on different cell types (K562, IMR90) test Hi-C data. 'IS' represents Insulation score and $^+$ indicates that diffusion augmentation is applied. **Bold**: best, underline: second best.: second best.

| Method | K562 | | | IMR90 | | |
|---|---|---|---|---|---|---|
| | SSIM | HiCRep | IS ($\downarrow$) | SSIM | HiCRep | IS ($\downarrow$) |
| Low-resolution | 0.232 | 0.860 | 7.284 | 0.349 | 0.944 | 5.724 |
| HiCPlus (Zhang et al., 2018) | 0.366 | 0.906 | 5.991 | 0.475 | 0.961 | 4.848 |
| HiCSR (Dimmick, 2020) | 0.383 | 0.917 | 5.550 | 0.502 | 0.964 | 3.892 |
| DeepHiC (Hong et al., 2020) | 0.370 | 0.919 | 6.843 | 0.485 | 0.965 | 4.697 |
| VEHiCLE (Highsmith & Cheng, 2021) | 0.444 | 0.908 | 6.100 | 0.574 | 0.958 | 4.622 |
| HiCBridge (ours) | 0.468 | 0.927 | 5.632 | 0.575 | 0.970 | 3.737 |
| HiCBridge$^+$ (ours) | **0.477** | **0.943** | **5.319** | **0.602** | **0.980** | **3.594** |

## 4.5 GENERALIZATION IN DIFFERENT RESOLUTIONS AND CELL TYPES

**Different resolutions.** We analysed the generalization performance of our model to assess its applicability in different resolutions. We compared SSIM as a vision metric, HiCRep as a Hi-C reproducibility, and Insulation score as a downstream task for data downsampled to specific ratios (1/16, 1/50, and 1/100) on GM12878 chromosome 20. As shown in Table 4, the results demonstrate the robust performance of our framework across varied resolutions. It is worth noting that the

downsampled data was obtained from high-resolution data, so the HiCrep and Insulation scores already indicate that downsampled data are similar to the high-resolution. We observed that for other models, except ours, the HiCRep and Insulation score are actually worse than the downsampled data. Furthermore, the lower the downsampling ratio, the less effective the augmentation becomes. Downsampled low-resolution data are visualized in Appendix F.

**Different cell types.** We compared the generalization of the model for different cell types: K562, IMR90. We employed SSIM, HiCRep and Insulation score for chromosomes 4, 14, 16, and 20 for each cell. From Table 5, we can observed that all models maintain their metric-specific performance on the GM12878 test data. For GM12878, VEHiCLE scored the highest of any previous model in visual acuity metrics, DeepHiC in HiCRep, and HiCSR in Insulation score. As with the GM12878 test data, we verified that our model also generalized better to other cell types. Furthermore, we verified that augmentation with a diffusion model helped to increase performance in various cell types. All of these results indicate that our framework is the most suitable model for real-world Hi-C data applications. We visualized insulation vectors which used for calculating Insulation score in IMR90 cell line in Appendix A.3, which confirms the accuracy of our method. The results for each model on different cell types are displayed in Appendix F.

## 4.6 ABLATION STUDY

Table 6: Comparison the regression and augmentation effect for GM12878 chromosome 20 based on generation method. **Bold**: best.

| Method | PCC | SCC | PSNR | SSIM | MSE ($\downarrow$) | SNR |
|---|---|---|---|---|---|---|
| Diffusion | 0.981 | 0.912 | 28.00 | 0.647 | 0.00164 | 783.5 |
| HiCBridge | 0.987 | 0.929 | 30.12 | 0.795 | 0.00100 | 999.0 |
| HiCBridge + DDB Aug. | 0.987 | 0.929 | 30.19 | 0.800 | 0.00099 | 1007.6 |
| HiCBridge + Diffusion Aug. | **0.988** | **0.930** | **30.22** | **0.802** | **0.00098** | **1011.3** |

**Regression with a diffusion model.** One could also obtain high-resolution Hi-C data from low-resolution with a diffusion model using Algorithm 1 and Algorithm 2 after replacing $x_1$ and $x_0$. To confirm the optimality of HiCBridge over this alternative, we trained the diffusion model on the same dataset and architecture. Table 6 demonstrates that HiCBridge has superior denoising performance compared to the diffusion model. Furthermore, HiCBridge takes only one step, resulting in a short inference time, unlike diffusion model that requires hundreds to thousands of steps.

**Augmentation with DDB.** In this paper, we generated low-resolution Hi-C data based on the diffusion model. Instead, by swapping inputs and outputs, one could use the DDB to generate new Hi-C data by learning to convert high-resolution Hi-C data to low-resolution data. To compare the alternative augmentation performance, we used HiCBridge as a baseline and compared the vision metrics with the alternative data augmentation scheme. From Table 6, we verified that HiCBridge$^+$, employing the diffusion model, yielded better results compared to the alternative augmentation method. Since DDB learns the expectation of the conditioned output, generating low-resolution data is relatively difficult. In contrast, the diffusion model is more suitable for generating low-resolution data due to its ability to estimate the distribution.

## 5 CONCLUSION

In this paper, we propose Hi-C enhancement using Direct Diffusion Bridge (HiCBridge) framework and its data augmented version (HiCBridge$^+$) for effectively enhance Hi-C resolution by cooperating with generation ability of diffusion model. Our comprehensive evaluations have shown that HiCBridge outperforms existing deep learning-based models on standard vision metrics, even without the need for additional specialized losses to boost performance. Our framework shows outstanding reproducibility in Hi-C analysis of human cells, particularly in tasks such as 3D structure matching, loop position reconstruction, and recovery of TAD boundaries. These results highlight the versatility and reliability of HiCBridge$^+$ at different resolutions and in different human cell types, and show that our model can be a promising tool for advancing chromatin research.

## 6 REPRODUCIBILITY STATEMENT

To ensure reproducibility, we provide detailed descriptions of our experiments in Appendix A, which is organized into three main sections. In Appendix A.1, we detail the source of the raw Hi-C data used in the experiments and outline the pre-processing methods employed to convert raw Hi-C data into contact matrices. Appendix A.2 contains the source code for the models and the hyperparameters setting for each model. Lastly, Appendix A.3 involves the various metrics we utilized to compare the performance of the models.

## 7 ETHICS STATEMENT

We have thoroughly read the ICLR Code of Ethics and affirm that this paper follows it. We will release the source code of our experiments and specify which program was used. Including our model, any model that converts low-resolution Hi-C data to high-resolution may potentially contain inaccurate biological information. To mitigate potential misinformation, it is crucial to thoroughly evaluate their applicability through diverse assessment methods.

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

# A  EXPERIMENTAL DETAILS

## A.1  HI-C DATA AVAILABILITY

**Download Hi-C data.** For GM12878 cell line, we used `GSE63525_GM12878_insitu_primary_30.hic` for high-resolution and `GSM1551550_HIC001_30.hic` for low-resolution Hi-C data. For K562, we used `GSE63525_K562_combined_30.hic` for high-resolution and `GSM1551620_HIC071_30.hic` for low-resolution. For IMR90, we used `GSE63525_IMR90_combined_30.hic` and `GSM1551602_HIC053_30.hic` for high-resolution and low-resolution data, respectively. We partitioned the dataset with chromosomes 1, 3, 5, 6, 7, 9, 11, 12, 13, 15, 17, 18, 19, and 21 as training set, chromosome 2, 8, 10, and 22 as validation set, and chromosome 4, 14, 16, and 20 as test set. Synthetic low-resolution Hi-C matrices were generated using the downsampling method introduced in Zhang et al. (2018).

**Pre-processing.** We obtained all Hi-C data from the Gene Expression Omnibus (GEO) GSE63525. We downloaded Hi-C data with mapping quality>30 and the Hi-C data were KR-normalized (Knight & Ruiz, 2013) to 10kb resolution using the Juicer software (Durand et al., 2016), resulting in Hi-C contact matrices. Zero values along the diagonal of the contact matrix were removed. We then threshold $99.9^{th}$ percentile and normalize value ranging from 0 to 1. The contact matrix was cropped to a size of $256 \times 256$ along the diagonal, except for VAE in VEHiCLE, for which the Hi-C contact matrix was cropped to a size of $244 \times 244$. Similar to previous works, we partitioned the contact matrix by overlapping every 50 bins to include contiguous information. Low-resolution Hi-C matrices were synthesized by downsampling high-resolution Hi-C data to specific resolutions (1/16, 1/50, and 1/100) following Liu & Wang (2019). We also pre-processed and generate Hi-C contact matrix of other human cell types (K562, IMR90) using the same process, as described above.

## A.2  MODEL DETAILS

Table 7: Comparative analysis of computational efficiency.

| Model | #Params | Inference Time (ms) |
|---|---|---|
| HiCPlus | 0.93k | 29 |
| HiCSR | 5.31M | 1708 |
| DeepHiC | 1.56M | 818 |
| VEHiCLE | 25.9M | 24 |
| HiCBridge | 20.6M | 158 |
| Diffusion Aug. | 20.6M | 6750 |

All of models were trained with GM12878 training set, encompassing a total of 3310 Hi-C contact matrix pairs sized $256 \times 256$. Our implementation was built with Pytorch (Paszke et al., 2019) and trained on a NVIDIA GeForce RTX 3090. All of models used Adam optimizer (Kingma & Ba, 2014). As shown in Table 7, we provide an overview of the computational efficiency of each model, comparing the number of model parameters and inference time per $256 \times 256$ Hi-C contact matrix. Note that VEHiCLE and HiCBridge are faster because they do not need to merge the cropped output, owing to their larger input image size.

**HiCPlus.** The model architecture from `https://github.com/wangjuan001/hicplus` was used. The model was optimized using the Mean Squared Error (MSE) loss and a learning rate of 3e-5. Training was performed with a batch size of 512 for 300 epochs.

**HiCSR.** The model architecture from `https://github.com/PSI-Lab/HiCSR` was used. We first trained the autoencoder with MSE loss with a learning rate of 1e-4 and a batch size of 4096 for 600 epochs. Subsequently, the GAN was optimized with GAN loss and feature reconstruction loss with a learning rate of 1e-5 for the generator and discriminator. We trained HiCSR with batch size of 8 for 500 epochs.

**DeepHiC.** The model architecture from `https://github.com/omegahh/DeepHiC` was used. The model was optimized with GAN loss, perceptual loss, total variation loss, and MSE

loss with a learning rate of 1e-4 for the generator and discriminator. We trained model with batch size of 16 for 200 epochs.

**VEHiCLE.** We used the model architecture from `https://github.com/Max-Highsmith/\VEHiCLE`. First, we trained VAE unsupervised with learning rate 1e-5, and a batch size of 512 for 50 epochs. The GAN was then optimized with GAN loss, Insulation loss, MSE loss, and VAE loss with a learning rate of 1e-5 for the generator and discriminator. Training for VEHiCLE was conducted with a batch size of 1 for 50 epochs.

**HiCBridge.** We used the U-net (Ronneberger et al., 2015) architecture from `https://github.com/lucidrains/denoising-diffusion-pytorch`. The details of architecture is provided in Table 8. When training our model, we sampled intermediate Hi-C matrices using (6) with $\delta = 1/1000$, $\epsilon = 0.01$ with Brownian motion. We then obtained the reconstructed Hi-C matrices in one-step without noise. Our model was optimized with $L_1$ loss with a learning rate of 1e-4. Training for HiCBridge was conducted with a batch size of 12 for 300 epochs.

**HiCBridge$^+$.** We trained a diffusion model to generate new low-resolution Hi-C data. Since the generated low-resolution data should correspond to the high-resolution, we conditioned the input of the diffusion model by concatenating the high-resolution data. Diffusion model was optimized with $L_1$ loss using the Adam optimizer with learning rate of 1e-4. Training for diffusion model was conducted with a batch size of 4 for 500 epochs. The details of architecture is provided in Table 8. Once the diffusion model was trained, we converted the high-resolution of training data to low-resolution using diffusion model. We then trained HiCBridge$^+$ using same architecture in HiCBridge. To match the number of data trained, we halved the epochs to 150 since the volume of dataset doubled. We trained HiCBridge$^+$ with $L_1$ loss with a learning rate 1e-4 and batch size of 4.

Table 8: Details of HiCBridge and diffusion model architecture.

|  | HiCBridge | Diffusion model |
|---|---|---|
| init_dim | 64 | |
| dim_mults | (1,1,2,2,4,4) | |
| channel | 1 | |
| loss_type | $L_1$ | |
| condition | False | True |
| noise_schedule | brownian | linear |
| objective | 'pred_x0' | 'pred_noise' |
| timesteps | 1000 | 1000 |
| inference steps | 1 | 500 |

## A.3 EVALUATION METHODS

**Standard Visual metrics.** We employed Scipy (Virtanen et al., 2020) module to measure Pearson correlation coefficient (PCC) and Spearmans correlation coefficient (SPC). We calculated peak signal-to-noise ratio (PSNR), structural similarity index metric (SSIM), mean squared error (MSE), and signal-to-noise ratio (SNR) using (8-11) with $C_1 = 0.01^2$ and $C_2 = 0.03^2$. $x$ and $y$ represent high-resolution Hi-C contact matrix and corresponding enhanced Hi-C data. $\mu_x$ and $\mu_y$ denote the mean of $x$ and $y$. $\sigma_x$ and $\sigma_y$ denote standard deviation of $x$ and $y$. $\sigma_{xy}$ represents covariance of $x$ and $y$.

$$PSNR(x, y) = 20 \log_{10} \left( \frac{\max(y) - \min(y)}{\sqrt{MSE(x, y)}} \right) \tag{8}$$

$$SSIM(x, y) = \frac{(2\mu_x \mu_y + C_1) + (2\sigma_{xy} + C_2)}{(\mu_x^2 + \mu_y^2 + C_1)(\sigma_x^2 + \sigma_y^2 + C_2)} \tag{9}$$

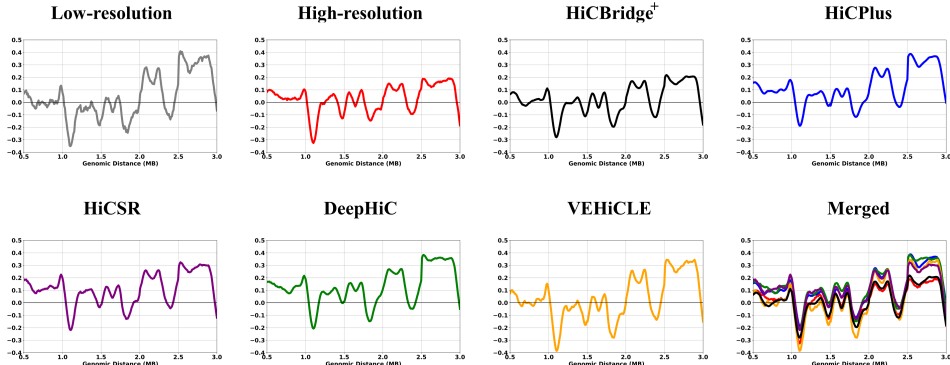

Figure 5: Insulation vector in IMR90 chromosome 4 0.5-3Mb Hi-C data. Merged represents over-lapped insulation vectors excluding low-resolution Hi-C data.

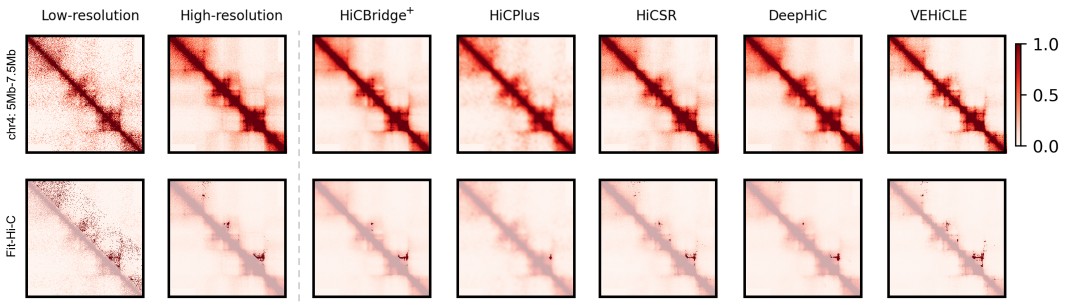

Figure 6: Visualization of important chromatin loop in GM12878 chromosome 4 5-7.5Mb via Fit-Hi-C.

$$MSE(x, y) = \sum_i (x_i - y_i)^2 \tag{10}$$

$$SNR(x, y) = \frac{\sum_i y_i}{\sqrt{\sum_i (x_i - y_i)^2}} \tag{11}$$

**GenomeDISCO and HiCRep.** In order to evaluate the Hi-C reproducibility, we converted the model outputs into a Hi-C contact matrix for the corresponding resolution. We then obtained the GenomeDISCO through the GenomeDISCO repository from `https://github.com/kundajelab/genomedisco` and the HiCRep through the HiCRep R package from `https://github.com/TaoYang-dev/hicrep`. We modified GenomeDISCO to be compatible with Python 3 version. We used default parameter in GenomeDISCO and $h = 20$ in HiCRep.

**TM-Score.** First, we employed 3D Max from `https://github.com/BDM-Lab/3DMax` to obtain `.pdb` file of Hi-C contact matrix. We only acquired 3D modeling at 250 bin size intervals. We then compared TM-Score via `https://github.com/Dapid/tmscoring`. We used convert factor as 0.6, reproducing number as 3, learning rate as 1, and maximum iteration as 1e4 in 3D Max.

**Insulation score.** We followed Crane et al. (2015) with window size as 20 bins. Fig. 5 represents insulation vector in IMR90 chromosome 4 using the aforementioned method.

**Jaccard Index.** We obtained significant intra-chromatin loop location using Fit-Hi-C from `https://github.com/ay-lab/fithic`. We then filtered loops for conditions where the q-value was lower than 1e-6 and the position of the loop was between 20e3 and 1e6 in genomic distance. With (12), we measured the Jaccard index for the high-resolution chromatin loop location. $x$ and $y$ denotes loop location of reconstructed Hi-C data and high-resolution Hi-C data, respectively. The comparison results are illustrated in Fig. 6

$$Jaccard(x, y) = \frac{|x \cap y|}{|x \cup y|} \tag{12}$$

# B ABLATION STUDY

**Effects of step size during training.** To verify that the performance improvement is due to a change in the structure of the model, we compared the performance of HiCBridge with one-step feed forward supervised learning by varying the step size during training. Notably, the step size of one is the same as the traditional supervised learning. From Table 9, we observed that increasing the step size during training leads to higher performance. This observation highlights that the HiCBridge framework, which can learn mappings with intermediate steps, is more efficient at learning the mappings from low-resolution to high-resolution Hi-C data.

**Effects of noise distribution.** In (6), we explored different noise distribution options to consider perturbation during measurement $x_1$. In order to compare the effect of noise schedule, we trained HiCBridge with three noise options: zero noise, noise fixed at 0.01, and the Brownian motion. From Table 10, We observed that the Brownian noise schedule achieved the best performance. Therefore, we applied the Brownian noise schedule when training HiCBridge and HiCBridge$^+$.

**Effects of augmentation ratio.** To validate results associated with data bias, we conducted a comprehensive performance analysis, examining the impact of the ratio of data augmented with the diffusion model to actual low-resolution data. For a fair comparison, we trained models with same training dataset and an identical amount of training data. Table 11 represents the results of standard visual metrics on GM12878 and IMR90 cell. We observed that a higher ratio of augmented data correlates with improved performance for the same cell type, yet diminished performance for other cell types. Notably, augmented data consistently contributes to enhancing the generalizability of HiCBridge.

Table 9: The results of standard vision metrics according to step size at training on GM12878 chromosome 20 Hi-C data. **Bold**: best.

| Step size $1/\delta$ | PCC | SCC | PSNR | SSIM | MSE ($\downarrow$) | SNR |
|---|---|---|---|---|---|---|
| 1 | 0.955 | 0.843 | 24.83 | 0.609 | 0.00353 | 547.7 |
| 5 | 0.980 | 0.915 | 28.21 | 0.753 | 0.00158 | 804.6 |
| 10 | 0.985 | 0.925 | 29.40 | 0.790 | 0.00119 | 919.8 |
| 100 | **0.987** | **0.929** | 30.00 | **0.798** | 0.00103 | 985.2 |
| 1000 | **0.987** | **0.929** | **30.12** | 0.795 | **0.00100** | **999.0** |

Table 10: The results of standard vision metrics according to noise distribution on GM12878 chromosome 4 Hi-C data. **Bold**: best.

| $\epsilon_t$ | PCC | SCC | PSNR | SSIM | MSE ($\downarrow$) | SNR |
|---|---|---|---|---|---|---|
| 0 | 0.956 | 0.887 | 21.32 | 0.441 | 0.00765 | 756.7 |
| 0.01 | 0.956 | 0.889 | 21.48 | 0.453 | 0.00737 | 771.0 |
| Brownian | **0.957** | **0.890** | **21.53** | **0.455** | **0.00728** | **775.4** |

Table 11: Comparative results of data augmentation ration on chromosome 4. **Bold**: best.

| Authentic : synthesized | PCC | SCC | PSNR | SSIM | MSE ($\downarrow$) | SNR |
|---|---|---|---|---|---|---|
| *GM12878* | | | | | | |
| 1 : 0 (HiCBridge) | 0.957 | 0.890 | 21.54 | 0.455 | 0.00727 | 776.4 |
| 1 : 0.5 | 0.958 | **0.891** | 21.91 | 0.457 | 0.00670 | 810.3 |
| 1 : 1 (HiCBridge$^+$) | **0.959** | **0.891** | 21.86 | **0.458** | 0.00678 | 805.6 |
| 0.5 : 1 | **0.959** | **0.891** | **21.95** | **0.458** | **0.00663** | **815.0** |
| *IMR90* | | | | | | |
| 1 : 0 (HiCBridge) | 0.927 | 0.845 | 18.99 | 0.380 | 0.01289 | 554.5 |
| 1 : 0.5 | **0.931** | 0.846 | **19.60** | 0.386 | **0.01125** | **594.7** |
| 1 : 1 (HiCBridge$^+$) | **0.931** | **0.847** | 19.52 | **0.392** | 0.01144 | 589.7 |
| 0.5 : 1 | 0.930 | 0.846 | 19.47 | 0.383 | 0.01159 | 585.7 |

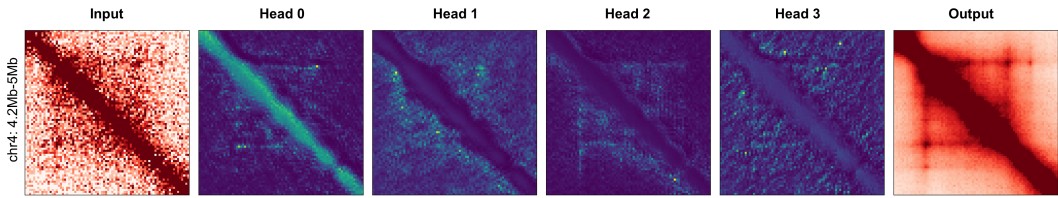

Figure 7: Visualized the attention map of the self-attention module in HiCBridge$^+$ on GM12878 chromosome 4 4.2Mb-5Mb.

## C  LIMITATION

It is important to note that our model is exclusively trained on inter-chromosomal connections, so a different method is needed to handle contact maps between chromosomes. Additionally, our models are limited to resolution enhancement for a bin size of 10kb. Lastly, biased data augmentation can potentially degrade the generalization of the model, as verified in Table 11, where the results of our model are shown to be influenced by data bias.

## D  INTERPRETABILITY

To understand how the model handles a given input, we analyzed the attention map of the self-attention module in the first layer of HiCBridge$^+$. As depicted in Fig. 7, we observed that each head emphasized a distinct region: head 0 allocates more attention to diagonal bins, head 1 highlights regions slightly off the diagonals, head 2 is dedicated to around 0.5Mb range of bins, and head 3 focuses on bins in more distant regions. Accordingly, our network is able to understand semantic structure of the contact map, a desired property for Hi-C data processing.

## E  GENERALIZABILITY

We conducted a series of experiments to elucidate the generalization abilities of our models. We evaluated standard visual metrics, HiCRep and TAD Insulation score on different resolution (GM12878 from GSM1551551), different cell type (HMEC from GSE63525 and GSM1551610) and another species (CH12-LX from GSE63525 and GSM1551640). In the Table 12, our models also reconstructed visual information and TAD boundaries on all Hi-C datasets. HMEC, with a resolution of about 1/20, poses a noisier dataset than our training data, and CH12-LX, being a mouse lymphoma cell, introduces a cross-species challenge. We supposed that these data shifts might contribute to the variation in SSIM or HiCRep performance in our models.

Table 12: Quantitative evaluation on another GM12878, HMEC and CH12-LX chromosome 4 Hi-C data. **Bold**: best.

| Method | Standard Visual Metrics | | | | | | Reproducibility | Downstream |
|---|---|---|---|---|---|---|---|---|
| | PCC | SPC | PSNR | SSIM | MSE ($\downarrow$) | SNR | HiCRep | IS ($\downarrow$) |
| *GM12878$_{GSM1551551}$* | | | | | | | | |
| Low-resolution | 0.807 | 0.680 | 14.40 | 0.292 | 0.03750 | 339.3 | **0.981** | 4.731 |
| HiCBridge$^+$ | **0.962** | **0.897** | **22.33** | **0.480** | **0.00606** | **846.6** | 0.980 | **2.578** |
| *HMEC* | | | | | | | | |
| Low-resolution | 0.476 | 0.397 | 9.62 | **0.123** | 0.10968 | 172.3 | **0.850** | 16.253 |
| HiCBridge$^+$ | **0.791** | **0.690** | **13.16** | 0.114 | **0.04894** | **259.5** | 0.801 | **11.344** |
| *CH12-LX* | | | | | | | | |
| Low-resolution | 0.690 | 0.585 | 13.09 | **0.324** | 0.05035 | 207.8 | **0.953** | 10.417 |
| HiCBridge$^+$ | **0.858** | **0.709** | **17.12** | 0.281 | **0.02070** | **336.7** | 0.852 | **6.933** |

## F  FURTHER EXPERIMENTAL RESULTS

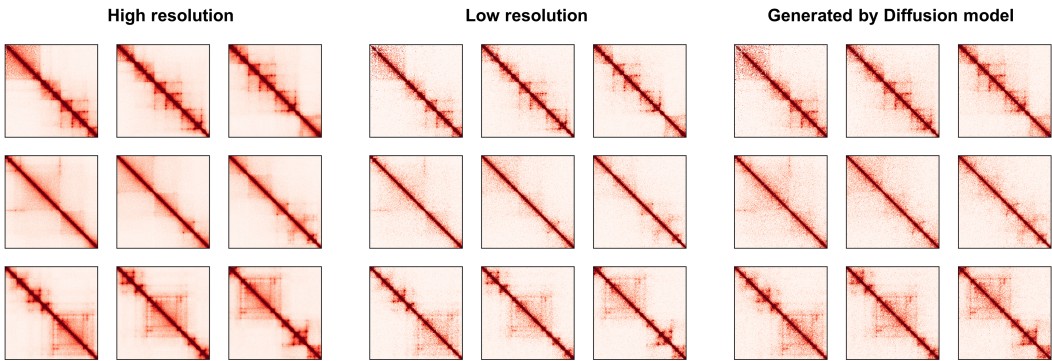

Figure 8: Visualized Hi-C data of GM12878 chromosome 13.

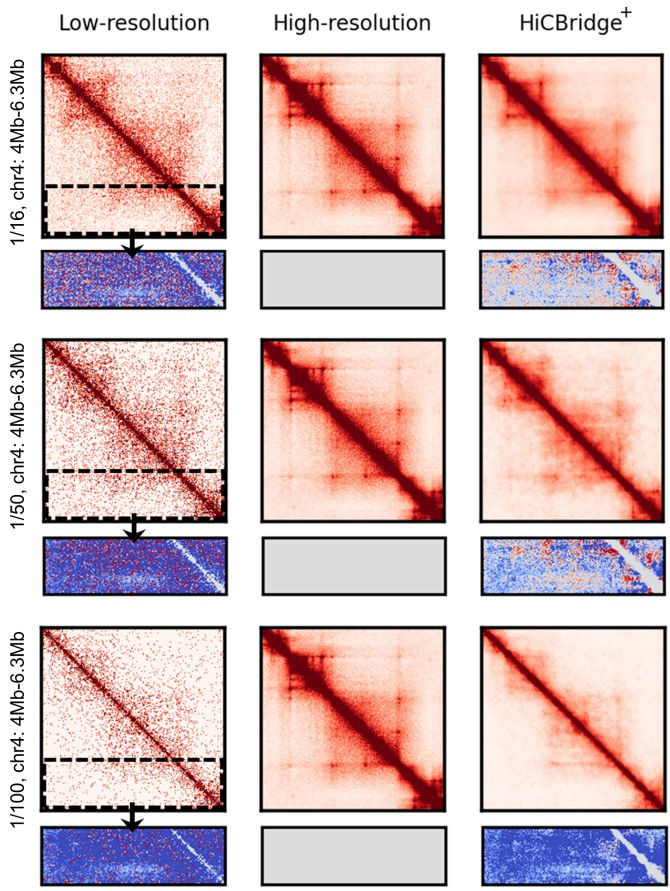

Figure 9: Visualization of Hi-C contact matrices on downsampling ratio 1/16, 1/50 and 1/100 in GM12878 chromosome 4 4-6.3Mb.

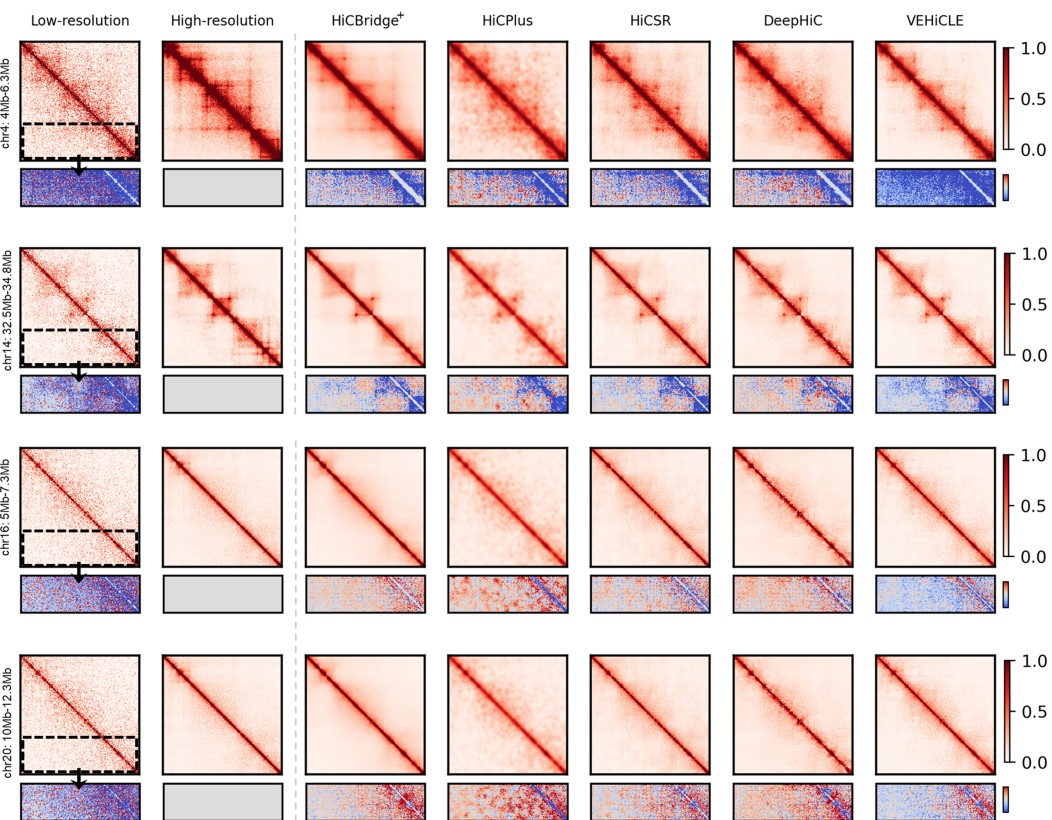

Figure 10: Visualization of Hi-C contact matrices in K562 cell line.

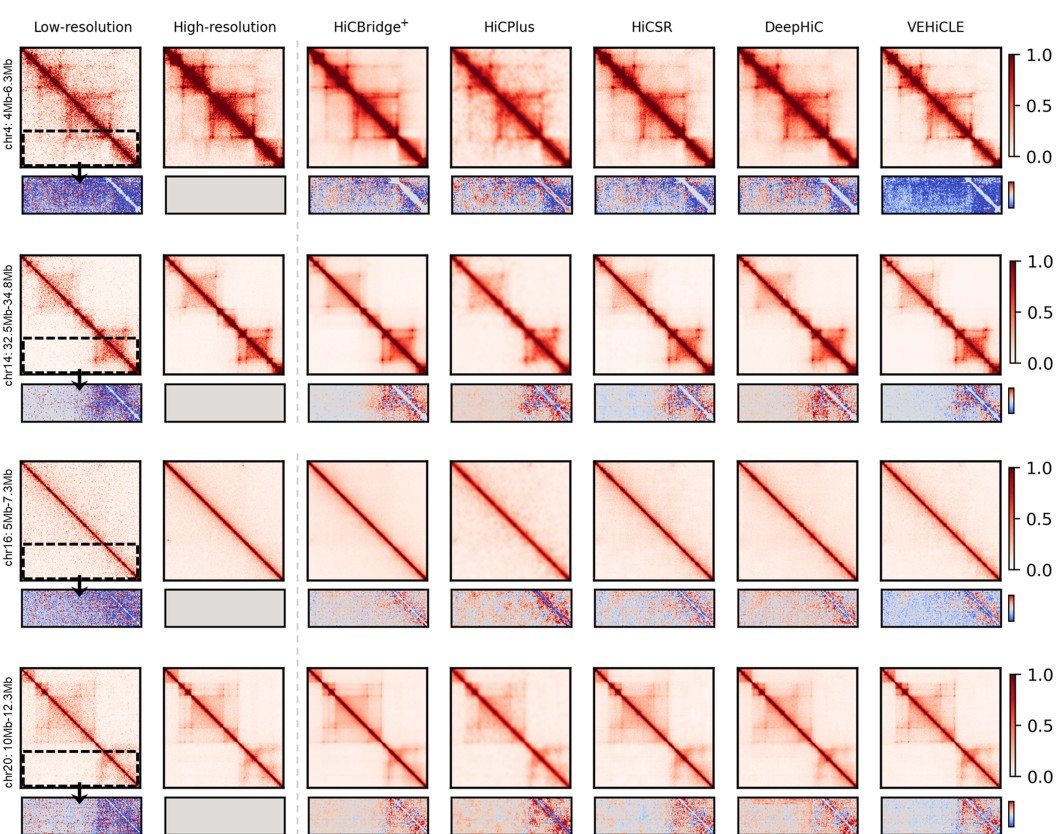

Figure 11: Visualization of Hi-C contact matrices in IMR90 cell line.

