# OpenReview forum: "HiCBridge: Resolution Enhancement of Hi-C Data Using Direct Diffusion Bridge"
_ICLR.cc/2024/Conference — Submitted to ICLR 2024_

### Official Review · Reviewer_eQMG · 2023-11-04

**Soundness:** 3 good
**Presentation:** 2 fair
**Contribution:** 3 good
**Rating:** 6
**Confidence:** 4

**Summary:**

The paper introduces HiCBridge, a new method for improving the resolution of Hi-C genomic data. HiCBridge utilizes a direct diffusion process to overcome the limitations of previous techniques, yielding higher quality data. It outperforms existing models in accuracy and reliability, offering a robust tool for genomic research.

**Strengths:**

The paper introduces an original technique leveraging a direct diffusion bridge to enhance Hi-C data resolution, showcasing creativity in addressing the limitations of existing deep learning methods. The quality of research is evident through rigorous evaluation against standard metrics and the demonstration of reproducibility, indicative of robust experimental design. Clarity is a strength of the paper, with concise explanations of complex methodologies and clear communication of results. The significance of the work is considerable, offering a tool with potential for broad applications in genomic research, with implications for understanding genomic architecture and influencing disease research. Overall, the paper represents a meaningful advance in bioinformatics, combining originality and quality with clear presentation and significant potential impact.

**Weaknesses:**

The methodology underpinning HiCBridge, while innovative, could be more transparent, particularly in how it fits within broader Hi-C analysis workflows and its robustness to data biases and hyperparameter variations. A thorough comparative analysis against a wider array of both deep learning and traditional approaches could better contextualize its performance claims. The validation of HiCBridge could also be strengthened by testing across diverse Hi-C datasets to ensure its generalizability. Furthermore, elucidating the biological significance of the resolution enhancement through detailed case studies would demonstrate the practical impact of the method. The paper would benefit from a discussion on computational efficiency, an essential consideration for large-scale genomic data analysis. Lastly, an in-depth consideration of the method's limitations, such as the effects of sequencing depth and data noise, would provide a more balanced view and inform future enhancements. Addressing these areas could significantly solidify the paper's contributions and utility in the field.

**Questions:**

What are the potential limitations of HiCBridge, particularly concerning sequencing depth, input data resolution, and noise in Hi-C datasets?
How might these limitations affect the application of HiCBridge in different research scenarios?

---

> ### Author Response · Authors · 2023-11-18
> **Reply to Reviewer eQMG**
>
> We thank the reviewer for the constructive and valuable comments. Please see the detailed response below.
>
> > Weaknesses 1. Show its robustness to data biases and hyperparameter variations.
>
> Please see **General Comments 3. Robustness to data biases**. We compared the ratio of data augmentation, as a hyperparameter, to verified the robustness of HiCBridge$^+$. While augmented data is beneficial for the generalization of HiCBridge, a high proportion of augmented data can pose challenges, contingent on data biases.
>
>
> > Weaknesses 2. A thorough comparative analysis against a wider array of both deep learning and traditional approaches. And by testing across diverse Hi-C datasets to ensure its generalizability.
>
> We obtained enhanced Hi-C contact matices by Gaussian smoothing as a traditional method. We employed 2D Gaussian distribution with kernel size $n = 17$ and $\sigma_x = \sigma_y = 3$. The table below reveals that, although the traditional method demonstrates competitive results, our approach consistently outperforms all matrices. These findings underscore the superiority of our method.
>
> |Method|PCC|SPC|PSNR|SSIM|MSE|SNR|HiCRep|Insulation score|
> |---|---|---|---|---|---|---|:---:|:---:|
> |Low resolution|0.758|0.630|13.24|0.234|0.04848|297.2|0.917|7.967|
> |Gaussian smoothing|0.948|0.875|18.31|0.348|0.01498|531.2|0.977|7.639|
> |HiCBridge$^+$|0.959|0.891|21.86|0.458|0.00678|805.6|0.984|3.365|
>
> > Weaknesses 3. Elucidating the biological significance of the resolution enhancement through detailed case studies.
>
> Please see **General Comments, 4. Biological Significance**. We compared our model with traditional MSE and GAN methods, and found that our model preserves structural information without introducing artifacts that can be misleading.
>
> > Weaknesses 4. Computational efficiency, an essential consideration for large-scale genomic data analysis.
>
> Please see **General Comments, 2. Computational Efficiency**. About 250 million lengths of chromosome 1 can be represented by about 430 $\times$ 256 $\times$ 256 Hi-C contact maps, which our model takes about 70 seconds to process.
>
> > Weaknesses 5. An in-depth consideration of the method's limitations, such as the effects of sequencing depth and data noise.
>
> It is important to note that our model is exclusively trained on inter-chromosomal connections, so a different method is needed to handle contact maps between chromosomes. Additionally, our models are limited to resolution enhancement for a bin size of 10kb. Lastly, biased data augmentation can potentially degrade the generalization of the model, as discussed in **Weakness 1**, where the results of our model are shown to be influenced by data bias. We have added these as our limitation in **Appendix C** of the revised paper.
>
> Finally, the reviewer is kindly reminded that the data noise is mostly originated from shallow sequence depth and one of the main goals of our study is to enhancement data quality from relatively shallow sequencing depth using direct diffusion bridge. The low-resolution Hi-C data we trained corresponds to a sequence depth of about 1/16 of the high-resolution data. In this paper, our model has shown robust performance for different resolutions (1/16, 1/50 and 1/100) and different cell types; K562 (about 1/14), IMR90 (about 1/10).

---

### Official Review · Reviewer_iBuE · 2023-11-04

**Soundness:** 3 good
**Presentation:** 3 good
**Contribution:** 2 fair
**Rating:** 6
**Confidence:** 3

**Summary:**

This paper proposes an Hi-C enhancement using Direct Diffusion Bridge (HiCBridge) model. The proposed method learns the mapping from low-resolution to high-resolution Hi-C data without experiencing mode collapsing issues commonly observed in Generative Adversarial Networks (GANs) or texture blurring that can occur in standard supervised deep learning approaches. The proposed model demonstrates good performance on standard vision metrics, various biological metrics, and downstream tasks across diverse cell types and resolutions.

**Strengths:**

+ As to the overall paper structure, I think it is clear and easy to follow.

+ The method used in the paper is reasonable. In simpler terms, the direct diffusion bridge allows us to calculate the likelihood of a diffusion process reaching a particular state at a specific time, without needing to consider the entire path the process took.

+ The experimental results underscore the versatility and reliability of HiCBridge+ across various resolutions and human cell types. These findings may have the potential to be a valuable tool for advancing chromatin research.

**Weaknesses:**

- The authors seem to have only used existing DDPM model to generate new low-resolution Hi-C data corresponding to the high-resolution ones. The authors have primarily relied on existing methods to generate the data, which, in itself, does not constitute a significant contribution. It is crucial for the authors to clearly explain the problem they aim to address and the challenges associated with it.

- The authors claim that the conventional regression models, which are based on the Mean Squared Error (MSE) loss, cause the model to regress to the mean in the target domain. This regression leads to blurring and loss of details. However, the author do not provide visual results to demonstrate this issue. The authors should elaborate on the challenges posed by the problem. Describing the difficulties and complexities associated with addressing the problem will highlight its importance and demonstrate the need for the proposed solutions.

-----------------------After Rebuttal---------------------------

Thank you for your feedback. The rebuttal addressed my concerns well. Considering other reviews, I have decided to increase my score.

**Questions:**

Please refer to the weakness part.

---

> ### Author Response · Authors · 2023-11-18
> **Reply to Reviewer iBuE**
>
> We thank the reviewer for your time and efforts in reviewing our paper, and your constructive comments. Please see the detailed response below.
>
> > Weaknesses 1. The authors seem to have only used existing DDPM model to generate new low-resolution Hi-C data corresponding to the high-resolution ones. The authors have primarily relied on existing methods to generate the data, which, in itself, does not constitute a significant contribution. It is crucial for the authors to clearly explain the problem they aim to address and the challenges associated with it.
>
> Please see **General Comments, 1. Difference from original methods**. We explored the best approach to leverage both the regression and generation capabilities of DDB and diffusion model. Various metrics show that our framework outperforms other models at Hi-C enhancement task. Additionally, our model has advantages not only in performance but also in the efficiency of Hi-C enhancement.
>
> > Weaknesses 2. The authors claim that the conventional regression models, which are based on the Mean Squared Error (MSE) loss, cause the model to regress to the mean in the target domain. This regression leads to blurring and loss of details. However, the author do not provide visual results to demonstrate this issue. The authors should elaborate on the challenges posed by the problem. Describing the difficulties and complexities associated with addressing the problem will highlight its importance and demonstrate the need for the proposed solutions.
>
> Please see **General Comments, 4. Biological Significance**. We highlight the limitations of models trained with traditional MSE and GAN approaches. Our model showcases conservative reconstruction, preserving structural information without introducing artifacts that could potentially lead to misinterpretation of the reconstructed Hi-C data.
>
> An MSE-based model tends to yield blurry output as it represents the average of all plausible reconstructions. In contrast, our approach breaks down the mapping into multiple timesteps, preventing regression to the mean. While GAN-based models excel at generating photo-realistic images by training generator and discriminator, a high level of similarity in the image domain does not necessarily ensure excellent biological reconstruction. Previous methods have incorporated various additional losses to involve biological meaning, but these approaches reduce the interpretability of the model and lead inadvertent results. Our method avoids using additional losses like perception loss or insulation loss.

---

### Official Review · Reviewer_CoBE · 2023-11-04

**Soundness:** 3 good
**Presentation:** 3 good
**Contribution:** 3 good
**Rating:** 6
**Confidence:** 4

**Summary:**

The paper propose to use Direct Diffusion Bridge to enhance the resolution of HI-C data. To promote the generalization in real-world situations, the authors introduce diffusion models to generate training data with more variation. Rich experimental results support excellent performance as well as its application to Hi-C analysis of human cells.

**Strengths:**

-	The paper proposes a diffusion based method to enhance resolution of HI-C data, which avoids over-smooth of standard supervised learning or training instability of GANs. The performance seems more competitive compared with previous methods.
-	The authors provide rich figures and tables to vividly and rigorously support their claims.
-	The paper is well-organized and easy to follow.

**Weaknesses:**

-	Although the application of diffusion-based methods to Hi-C is meaningful, the method seems lack of novelty. The authors should explain carefully the difference between the core method and previous Direct Diffusion Bridge.
-	I suggest the authors add comparison about memory and time consumption to help readers get a full picture of their method.

**Questions:**

-	Can you provide visualization of generated training data?
-	During the process of enhancing resolution, new content will sprout. I am concerned whether the method will generate inrelevant or even harmful content, which may incur severe results.

---

> ### Author Response · Authors · 2023-11-18
> **Reply to Reviewer CoBE**
>
> Thanks for the valuable and constructive comment. For point-to-point response, see below.
>
> > Weaknesses 1. Although the application of diffusion-based methods to Hi-C is meaningful, the method seems lack of novelty. The authors should explain carefully the difference between the core method and previous Direct Diffusion Bridge.
>
> Please see **General Comments, 1. Difference from original methods.** We propose the best framework for the Hi-C enhancement task through a combination of DDB and diffusion. Considering the regression performance of the DDB and the generative performance of the diffusion model at the same time, our model showed superior results on various metrics.
>
> > Weaknesses 2. I suggest the authors add comparison about memory and time consumption to help readers get a full picture of their method.
>
> Please see **General Comments, 2. Computation efficiency.** Our model takes 0.16 seconds to process a single 256 $\times$ 256 Hi-C contact map and has a competitive memory requirement.
>
>
> > Question 1. Can you provide visualization of generated training data?
>
> We visualized generated Hi-C data of GM12878 chromosome 13 using diffusion augmentation model in **Fig. 8 in Appendix F** of the revised paper. We used the generated data in order to train HiCBridge$^+$.
>
>
> > Question 2. During the process of enhancing resolution, new content will sprout. I am concerned whether the method will generate inrelevant or even harmful content, which may incur severe results.
>
> Please see **General Comments, 4. Biological significance.** We verified HiCBridge recovers structural information without misleading contents.

---

> > ### Comment · Reviewer_CoBE · 2023-11-22
> >
> > All my concerns have been addressed in the rebuttal. I have no further comment.

---

### Official Review · Reviewer_jkUa · 2023-11-05

**Soundness:** 3 good
**Presentation:** 3 good
**Contribution:** 3 good
**Rating:** 6
**Confidence:** 3

**Summary:**

This paper addresses the resolution limitations inherent in Hi-C analysis—a genomic technique that elucidates the three-dimensional architecture of chromosomes. Hi-C's utility in revealing critical genomic structures such as A/B compartments and chromatin loops is constrained by the resolution of the data, which refers to the granularity of the contact matrix generated from sequencing reads. The resolution challenge is compounded by the quadratic increase in sequencing efforts required for finer resolution, thus escalating the costs. Previous attempts to enhance the resolution of Hi-C data have leveraged deep learning approaches like HiCPlus and HiCNN; however, these methods are often limited by the computational demands and the quantity of high-resolution data needed for training.

In this context, the authors propose HiCBridge, a novel method that employs a Direct Diffusion Bridge (DDB) to learn the transformation from low to high-resolution Hi-C data. This method is designed to circumvent the shortcomings of conventional deep learning techniques that may result in overly smooth textures or mode collapse. HiCBridge integrates diffusion model-based data augmentation to handle a wider array of real-world data variations, thereby enhancing the model's applicability and robustness. The model's potential to bolster downstream genomic analyses—such as 3D structure matching and loop position reconstruction—positions it as a significant advancement in genomic research, offering a path to high-resolution Hi-C data without the prohibitive costs of extensive sequencing.

**Strengths:**

Originality
The submission presents a novel approach to enhancing the resolution of Hi-C data through the Direct Diffusion Bridge (DDB), distinguishing itself from prior work that primarily relies on conventional deep learning methods. The originality of the paper lies not just in the application of diffusion models—a relatively recent trend in machine learning—but in the specific formulation of using such models to bridge low-resolution and high-resolution Hi-C data. This creative combination of diffusion processes with genomic data analysis is novel, as it deviates from the usual convolutional neural network (CNN) approaches that dominate the field.

Quality
The paper conducts experiments on multiple standard datasets, and presents a good ablation analysis.

Clarity
The paper is well written. I find it easy to read, the figures along with the captions are appropriate and aid understanding of the paper.

**Weaknesses:**

1. The robustness of the model against various datasets and its performance under different conditions (such as varying levels of input resolution) would need thorough examination.
2. The paper would benefit from a more detailed comparison with state-of-the-art models. This includes not only performance metrics but also computational efficiency, scalability, and the amount of training data required.
3. The model's generalizability to different types of Hi-C data, such as those from various species or cells in different states, should be assessed. If the model has only been tested on a narrow range of data, its practical applicability could be limited.
4. The paper should address the interpretability of the HiCBridge model. Understanding how the model makes its predictions is crucial, particularly in genomic studies where the biological implications of the findings are as important as the findings themselves.

**Questions:**

see weaknesses

---

> ### Author Response · Authors · 2023-11-18
> **Reply to Reviewer jkUa**
>
> Thank you for your detailed feedback and the encouraging comments. Please see the detailed response below.
>
> > Weaknesses 1. The robustness of the model against various datasets and its performance under different conditions (such as varying levels of input resolution) would need thorough examination.
>
> We conducted a series of experiments to elucidate the generalization abilities of our models. We evaluated standard visual metrics, HiCRep and TAD Insulation score on different resolution (GM12878 from GSM1551551), different cell type (HMEC from GSE63525 and GSM1551610) and another species (CH12-LX from GSE63525 and GSM1551640). In the **Table 12 in Appendix E** of the revised paper, our models also reconstructed visual information and TAD boundaries on all Hi-C datasets. HMEC, with a resolution of about 1/20, poses a noisier dataset than our training data, and CH12-LX, being a mouse lymphoma cell, introduces a cross-species challenge. We supposed that these data shifts might contribute to the variation in SSIM or HiCRep performance in our models.
>
>
> |Method|PCC|SPC|PSNR|SSIM|MSE|SNR|HiCRep|Insulation score|
> |---|---|---|---|---|---|---|:---:|:---:|
> |GM12878$_{GSM1551551}$||||||||||
> |Low resolution|0.807|0.680|14.40|0.292|0.03750|339.3|0.981|4.731|
> |HiCBridge$^+$|0.962|0.897|22.33|0.480|0.00606|846.6|0.980|2.578|
> |HMEC||||||||||
> |Low resolution|0.476|0.397|9.62|0.123|0.10968|172.3|0.850|16.253|
> |HiCBridge$^+$|0.791|0.690|13.16|0.114|0.04894|259.5|0.801|11.344|
> |CH12_LX||||||||||
> |Low resolution|0.690|0.585|13.09|0.324|0.05035|207.8|0.953|10.417|
> |HiCBridge$^+$|0.858|0.709|17.12|0.281|0.02070|336.7|0.852|6.933|
>
> > Weaknesses 2. The paper would benefi from a more detailed comparison with state-of-the-art models. This includes not only performance metrics but also computational efficiency, scalability, and the amount of training data required.
>
> Please see **General Comments, 2. Computation efficiency**.
>
> > Weaknesses 3. The model's generalizability to different types of Hi-C data, such as those from various species or cells in different states, should be assessed. If the model has only been tested on a narrow range of data, its practical applicability could be limited.
>
> As explained in the response to In **Weakness 1**, we verified our model’s generalizability at different cell type (HMEC; Human Mammary Epithelial Cells) and different species (CH12-LX; Mouse Lymphoma Cells).
>
> > Weaknesses 4. The paper should address the interpretability of the HiCBridge model. Understanding how the model makes its predictions is crucial, particularly in genomic studies where the biological implications of the findings are as important as the findings themselves.
>
> To understand how the model handles a given input, we analyzed the attention map of the self-attention module in the first layer of HiCBridge$^+$. As depicted in **Fig.7 in Appendix D** of the revised paper, we observed that each head emphasized a distinct region: head 0 allocates more attention to diagonal bins, head 1 highlights regions slightly off the diagonals, head 2 is dedicated to a 0.5Mb range of bins, and head 3 focuses on bins in more distant regions. Accordingly, our network can understand semantic structure of the contact map, a desired property for Hi-C data processing.

---

### Official Review · Reviewer_LnZX · 2023-11-05

**Soundness:** 3 good
**Presentation:** 3 good
**Contribution:** 2 fair
**Rating:** 3
**Confidence:** 4

**Summary:**

This paper borrows idea from the Direct Diffusion Bridge that establishing the diffusion process between the clean and corruption for Hi-C data super-resolution. And a diffusion-based data augmentation method is proposed for adapting to the real-world situations. Extensive experiments demonstrate the effectiveness of the proposed method.

**Strengths:**

1. This paper introduces a diffusion-based data augmentation method to alleviate the real-world variation.
2. Extensive experiments on Hi-C data analysis demonstrate the effectiveness of the proposed method.

**Weaknesses:**

1. The contribution of this paper is limited. The proposed HiCBridge directly borrows idea from the Direct Diffusion Bridge without any improvement. While the proposed diffusion-based data augmentation (Algorithm 1 and 2) is also trivial. It seems like the paper tends to apply the existing method to the new task, but lack of novelty.
2. It is not clear how the Algorithm 1 can be used to generate the low-resolution data as your target distribution is based on x0 (high-resolution Hi-C data). Please check whether the x0 in Algorithm 1 should be x1.
3. It seems like the diffusion augmentation is unusable in higher resolution cases for inferior performance, according to the Table 4.

**Questions:**

lease refer to the paper weaknesses.

---

> ### Author Response · Authors · 2023-11-18
> **Reply to Reviewer LnZX**
>
> We would like to thank the reviewer for the constructive comments and the thorough feedback. For point-to-point response, see below.
>
> > Weaknesses 1. The contribution of this paper is limited. The proposed HiCBridge directly borrows idea from the Direct Diffusion Bridge without any improvement. While the proposed diffusion-based data augmentation (Algorithm 1 and 2) is also trivial. It seems like the paper tends to apply the existing method to the new task, but lack of novelty.
>
> Please see **General Comments, 1. Difference from original methods** along with the revised paper. We highlight that we found the optimal combination of DDB and diffusion for Hi-C enhancement task, which cannot be regarded as trivial given the importance of the Hi-C data processing.
>
> > Weaknesses 2. It is not clear how the Algorithm 1 can be used to generate the low-resolution data as your target distribution is based on x0 (high-resolution Hi-C data). Please check whether the x0 in Algorithm 1 should be x1.
>
> Thanks for pointing out the typo. We have fixed our Algorithm 1 accordingly.
>
> > Weaknesses 3. It seems like the diffusion augmentation is unusable in higher resolution cases for inferior performance, according to the Table 4.
>
> If our interpretation of the reviewer's comment is accurate, the aim is to clarify the subpar performance of HiCBridge$^+$ at extremely low resolutions (1/50 and 1/100). As highlighted in the **General Comments, 3. Robustness to data biases**, our analysis suggests that biased data augmentation holds the potential to compromise the model's generalization. While the incorporation of generated low-resolution data contributes to the overall generalization of HiCBridge, an excessive bias with such data may lead to a deterioration in HiCBridge$^+$'s performance across diverse Hi-C datasets. It is crucial to note that the data compared in Table 4 of our paper consists of downsampled data rather than real low-resolution data, presenting a distribution distinct from the one the model was trained on [1]. This finding offers a plausible explanation for the observed reduction in the generalizability of HiCBridge$^+$ compared to HiCBridge, especially at extremely low-resolution settings.
>
> [1] A comprehensive evaluation of generalizability of deep-learning based Hi-C resolution improvement methods, Murtaza et al., Biorxiv, 2022

---

> ### Author Response · Authors · 2023-11-21
> **Dear Reviewer LnZX**
>
> As the deadline for the Reviewer-Author discussion phase is fast approaching (there is only a day left), we respectfully ask whether we have addressed your questions and concerns adequately.

---

> ### Author Response · Authors · 2023-11-23
> **Dear Reviewer LnZX**
>
> Dear reviewer LnZX,
>
> This is a kind reminder as the deadline is imminent.  We believe that we have addressed the concerns that you have raised. Specifically,
>
> 1. We have distinguished our work from previous papers by finding a proper combination of diffusion model and direct diffusion bridge for Hi-C enhancement to establish an effective and efficient framework.
>
> 2. We have corrected Algorithm 1, which was incorrectly stated.
>
> 3. We have explained that HiCBridge$^+$ underperforms at extremely low resolutions. An experiment in **Table 11 in Appendix B** showed that diffusion augmentation can suffer performance degradation due to shifts in the data distribution.
>
> We would like to gently remind you that the end of the discussion period is imminent. We would appreciate it if you could let us know whether our comments addressed your concerns.
>
> Best regards, Authors

---

### Author Response · Authors · 2023-11-18
**General Reply to All Reviewers**

Modified contents have been highlighted red in the revised paper.

We would like to express our gratitude to the reviewers for their valuable and thorough feedback. We are encouraged by reviewers' comments such as "method to address real-world variation and extensive experiments" (LnZX), "originality, quality and clarity" (jkUa), "competitive, rigorous support, and well-organized" (CoBE), "clear, reasonable, and versatility" (iBuE), and "creativity, rigorous evaluation, clarity and significant potential impact" (eQMG).

After meticulously analyzing the received responses, we identified four recurrent categories of questions.

## 1. Difference from original methods (DDB)

Several reviewers (iBuE, CoBE and LnZX) have expressed curiosity regarding the distinctions between our approach and the traditional Direct Diffusion Bridge (DDB). In response, we conducted a comparative analysis of the DDB method and the diffusion model, evaluating their generation and denoising performance to identify the optimal combination for Hi-C enhancement.

Considering that both the diffusion model and DDB can be viewed as generative models, we explored their joint usage to maximize performance, as detailed in **Section 4.6 (Ablation study)** of the revised paper. The results confirm that our approach—employing data augmentation with a diffusion model followed by regression of low-resolution Hi-C data to high-resolution with DDB—outperforms other methods according to standard visual metrics while maintaining biological relevance. A fixed bin size (i.e. 10kb in our case) the low resolution data from low sequence count are noisier than high resolution data, so the corresponding data distribution in probability density function has fewer isolated mode. Therefore, a diffusion model can learn the low-resolution data distribution much more easily compared to the high-resolution data distribution with many isolated modes. After augmenting the data with the diffusion model, leveraging a DDB that connects the two data distributions makes it easier to enhance Hi-C data than learning from a Gaussian to a high-resolution data distribution.

Moreover, our approach not only perform well but has advantages specific to Hi-C data. Due to the large size of genomic data, a DDB that can perform Hi-C enhancement in one step is more efficient than a diffusion model that requires hundreds of timesteps for inference. In addition, low-resolution Hi-C data and high-resolution Hi-C data are highly similar in the image. We utilized this to reduce the time by perturbing with t=0.5 and then denoising for the remaining timesteps at diffusion augmentation.

## 2. Computational efficiency

All models were trained using the same dataset, encompassing a total of 3310 Hi-C contact matrix pairs sized 256 $\times$ 256 from the GM12878 cell line. Given the varied input image sizes for each model, preprocessing was executed accordingly. For instance, HiCPlus, capable of handling 40 $\times$ 40 images, was trained with 36 cropped contact matrices from a single contact matrix. As shown in **Table 7 in Appendix A.2** of the revised paper, we provide an overview of the computational efficiency of each model, comparing the number of model parameters and inference time per 256 $\times$ 256 Hi-C contact matrix. Note that VEHiCLE and HiCBridge are faster because they do not need to merge the cropped output, owing to their larger input image size.


## 3.Robustness to data biases

To validate results associated with data bias, we conducted a comprehensive performance analysis, examining the impact of the ratio of data augmented with the diffusion model to actual low-resolution data. For a fair comparison, we trained models with same training dataset and an identical amount of training data. **Table 11 in Appendix B** of the revised paper represents the results of standard visual metrics on GM12878 and IMR90 cells. We observed that a higher ratio of augmented data correlates with improved performance for the same cell type, yet diminished performance for other cell types. Notably, augmented data consistently contributes to enhancing the generalizability of HiCBridge.

## 4. Biological importance

Some reviewers (CoBE and eQMG) pointed out whether our model could identify biological significance. In the **Fig. 2** in the revised paper, we highlight the limitations of models trained with traditional MSE and GAN approaches. HiCPlus, an MSE-based model, fails to capture existing TAD regions present in high-resolution data (blue arrows), whereas the GAN-based model, HiCSCR, exhibits outliers in locations where they should not be (yellow arrow). Those artifacts could potentially lead to misinterpretation of the reconstructed Hi-C data. In contrast, our model showcases conservative reconstruction, preserving structural information without introducing artifacts that could potentially lead to misinterpretation of the reconstructed Hi-C data.

---

### Meta-Review · Area_Chair_3MNv · 2023-12-13

**Metareview:**

The author introduces HiCBridge, a novel method that employs a Direct Diffusion Bridge (DDB) to learn the transformation from low-resolution to high-resolution Hi-C data. The proposed approach learns to map from low-resolution to high-resolution Hi-C data without encountering the mode collapse issue common in Generative Adversarial Networks (GAN) or the texture blurring problem that may arise in standard supervised deep learning methods. The proposed model demonstrates good performance across standard visual metrics, various biological benchmarks, and downstream tasks across different cell types and resolutions. The main concern is the lack of innovation in methodological contribution, suggesting that the proposed method lacks novelty.

**Justification For Why Not Higher Score:**

One reviewer provided negative feedback, while four reviewers gave comments slightly above the acceptance threshold. The main concern is that the method proposed in the paper lacks sufficient innovation, with the benefits primarily arising from the application of existing methods to the current task.

**Justification For Why Not Lower Score:**

N/A

---

### Decision · Program_Chairs · 2024-01-16

Reject